# Methods Used and Application of the Mouse Grimace Scale in Biomedical Research 10 Years on: A Scoping Review

**DOI:** 10.3390/ani11030673

**Published:** 2021-03-03

**Authors:** Alexandra L. Whittaker, Yifan Liu, Timothy H. Barker

**Affiliations:** 1School of Animal and Veterinary Sciences, Roseworthy Campus, The University of Adelaide, Roseworthy 5371, Australia; a1682344@student.adelaide.edu.au; 2JBI, Faculty of Health and Medical Sciences, The University of Adelaide, Adelaide 5005, Australia; Timothy.barker@adelaide.edu.au

**Keywords:** mouse grimace scale, pain, validity, methods, reliability

## Abstract

**Simple Summary:**

The Mouse Grimace Scale (MGS) was developed as a tool for the measurement of pain in laboratory mice. There have been a number of studies focused on the technique’s validity in different models, and across pain types. With this new information, it is important that a review following systematic methodology is performed on these studies to summarise the methods used, the validity across model types, and the effects of external variables. In this review, we present all of the available evidence on the MGS, together with an indication of the extent of the evidence available for each parameter considered. This review will provide an increased strength of evidence to guide researchers, ethics committees, and policy makers on the use and application of the MGS in biomedical research.

**Abstract:**

The Mouse Grimace Scale (MGS) was developed 10 years ago as a method for assessing pain through the characterisation of changes in five facial features or action units. The strength of the technique is that it is proposed to be a measure of spontaneous or non-evoked pain. The time is opportune to map all of the research into the MGS, with a particular focus on the methods used and the technique’s utility across a range of mouse models. A comprehensive scoping review of the academic literature was performed. A total of 48 articles met our inclusion criteria and were included in this review. The MGS has been employed mainly in the evaluation of acute pain, particularly in the pain and neuroscience research fields. There has, however, been use of the technique in a wide range of fields, and based on limited study it does appear to have utility for pain assessment across a spectrum of animal models. Use of the method allows the detection of pain of a longer duration, up to a month post initial insult. There has been less use of the technique using real-time methods and this is an area in need of further research.

## 1. Introduction

Mice are commonly used as models for a range of conditions in biomedical research. This use is globally significant, with approximately 5.7M mice used in Europe alone [1]. Many of these models may result in pain or sickness arising either directly, or from other pathological processes. Furthermore, a range of husbandry or routine procedures also undertaken in vivaria may also cause pain or distress. Assessment of affective states in research animals is important to enable the implementation of humane endpoints, thus meeting ethical and legal responsibilities, as well as enhancing the translational validity of animal research. However, the assessment of animal emotion is challenging, tending to combine behavioural and physiological measures to provide a holistic assessment [2,3,4,5]. There has been comparatively more research focus on negative states such as pain, and as such available methods have undergone more extensive testing and validation across a wide range of study types.

Pain is an important issue in animal research for several reasons [6]. Firstly, in recent years, there has been growing concern about translational failures from animal studies to the clinic [7]. Numerous authors have levelled criticism at animal models of pain for being poorly predictive of the clinical scenario [8,9], based on issues such as the variability between animals and the relevance of the assay outcomes to the human pain experience [10]. This concern is not unique to pain research; pain commonly arises in other disease conditions and may be a target for novel therapeutics. Secondly, pain and its sequelae may influence the results obtained from animal model studies, affecting a range of physiological and immunological processes. This further impacts on the reliability and translatability of the results obtained from these studies [11,12,13]. Finally, pain presents a significant cost to animal welfare through the impact on individual animals. Therefore, the assessment of pain and the application of methods to mitigate its effects are needed to safeguard animal welfare and to conform to ethical requirements in biomedical research—for instance, the refinement aspect of the 3Rs [14]. This assists in addressing societal concerns around the use of animals in research.

One of the more commonly used assessment methods, suggested to be specific to pain, is the use of facial expression scoring or the so-called ‘grimace scales’ [15,16]. The idea behind using facial expressions as a readout for pain neurobiology came from human facial codification scales [17,18]. The Facial Action Codification System (FACS) allows the categorization of movements of the facial muscles. Specific combinations of movements lead to changes in discrete facial regions or “facial action units (FAU)”—for instance, the closing of the eyelids. Recognition of changes in these FAUs has been proposed to allow determination of emotional state [19]. Grimace scales were developed for non-human animals, with the goal of standardizing methods for different species. The original grimace scale was developed for mice by Langford and colleagues in 2010 [20], and validated through the application of a variety of preclinical pain assays. In this scale, changes in five facial action units are assessed to determine level of pain: (1) orbital tightening, (2) nose bulge, (3) cheek bulge, (4) ear position, and (5) whisker change. Grimace scale development in other species followed (see [16] for full history), as did further examination of the Mouse Grimace Scale (MGS) in a range of animal models and conditions.

There have been a number of reviews on grimace scales in a variety of species [6,16,21,22], but none that focussed solely on mice or used systematic methods to identify all studies where the MGS was utilised. Now, 10 years on from the publication of the original study, a comprehensive systematic assimilation of the evidence on the MGS is warranted. Since mice are the most commonly used mammal in biomedical research [23], and given the methodology used in this review, the review is limited to this species. In contrast to a systematic review and meta-analysis, scoping reviews are broader in scope and bring together all current evidence, regardless of quality [24]. They may also pave the way for future systematic reviews on a clearly defined question identified in the scoping review. Therefore, the aim of this scoping review was to identify all published studies on the MGS and assimilate the evidence based on features of the scale use, with a particular focus on the application of the technique across a range of animal models, the methods used, and the impact of external variables on validity and reliability. This review will provide an increased strength of evidence to guide researchers, ethics committees, and policy makers on the use and application of the MGS in biomedical research.

## 2. Methods

JBI’s methodology for conducting scoping reviews was followed in the conduct and reporting of this review [25]. A protocol for this review was not registered since common protocol databases (e.g., PROSPERO) do not accept scoping review protocols.

### 2.1. Search Strategy

The search strategy aimed to locate published studies in English. An initial limited search of Medline was undertaken to identify articles on the topic. The keywords contained in the titles and abstracts of relevant articles, and the index terms used to describe the articles, were used to develop a full search strategy for Medline via Pubmed using MeSH and free text terms. The search strategy was adapted for Scopus and Web of Science (including CAB abstracts) database searches. The three databases were searched in May 2020 using the developed search strategies (see Appendix A). The search was updated in October 2020. Key concepts used for searching were “mice” and “grimace scale”. Hand searching of reference lists was performed to identify additional studies. Studies published from database inception were eligible for inclusion. Publications were excluded electronically if they were conference abstracts with full study detail and results not available, or review articles.

### 2.2. Eligibility Criteria

Studies were included if they investigated the Mouse Grimace Scale in mice irrespective of age, sex, or strain. Studies that looked at a change in any number of facial action units but that did not report this as use of a ‘grimace scale’ were excluded. Studies that used the MGS and reported it as such but modified the method slightly were, however, eligible for inclusion. Only studies that investigated the MGS based on an understanding that this was a measure of pain were eligible—for example, a study using the MGS to assess positive emotion would have been ineligible for inclusion. All study designs were eligible for inclusion. Studies investigating new ways of collecting MGS data, for example, by automation techniques, were excluded. However, studies evaluating the objective nature of the test, for example those studies examining reliability between observers or institutions, were eligible for inclusion.

### 2.3. Study Selection

Following the search, all identified citations were collated and uploaded into EndNote X8.0.1 and duplicates removed. Potentially relevant studies were retrieved in full and their citation details imported into Covidence (Veritas Health Innovation, Melbourne, Australia). Titles were screened by one reviewer (A.L.W.) for assessment against the inclusion criteria for the review. Abstract and full text screening were performed by all authors (A.L.W., Y.L., T.H.B.) with two independent reviewers being required to certify the inclusion of each study. Disagreements that arose between the reviewers at each stage of the study selection process were resolved through discussion with the third reviewer.

### 2.4. Data Extraction

Data were extracted from the included studies by three independent reviewers (A.L.W., Y.L., T.H.B.) using an electronic form developed by the authors. All reviewers initially performed independent reviews of the same 3 studies [26,27,28] to pilot the extraction tool and check for data consistency. Following this, the remaining studies were allocated between the 3 reviewers, with each study being extracted by one reviewer. The distribution of the papers between the data extractors was done randomly. Only data directly relevant to the research question were extracted. All data extracted were reviewed by the authorship team to ensure completeness of extraction. Contact with study authors was undertaken where necessary to clarify findings or seek further information. In accordance with guidelines on scoping reviews [25], the goal of the review was to provide an overview of evidence on the MGS regardless of quality. Hence, methodological quality assessment of included studies was not undertaken.

## 3. Results

### 3.1. Study Characteristics

A total of 240 articles were retrieved. Six studies were retrieved through hand searching of the reference lists of included studies or forward citation searching. Following title and abstract screening, 59 articles were assigned for full-text retrieval, with 48 articles being included in the full-text review (Figure 1). The reason for the majority (*n* = 7) of the exclusions after full text review was due to the studies evaluating MGS automation methods, rather than pain in mice. The characteristics of the included studies are presented in Table 1. Observational studies were eligible for inclusion. However, the majority (92%) of the studies (*n* = 43) adopted an experimental study design, using the typical randomized controlled trial (RCT) design or pseudo-RCT design (where allocation to groups is systematic and not random). The remaining studies adopted a quasi-experimental design, such as using a pre-test, post-test repeated measures design with no group running in parallel. Since the first report of the MGS by Langford and colleagues in 2010, the number of publications investigating the method has grown considerably to a current approximately steady state rate of around six to nine publications per year, sustained over the last 5 years (Figure 2).

### 3.2. Animal Model Characteristics

Studies were allocated into three categories based on the types of interventions applied to the mice for subsequent grimace score measurement. The categories considered were (1) animal model, (2) husbandry/procedural, and (3) biological. Studies were categorised as utilising animal models if they used an animal model of a human condition likely to cause pain. Husbandry/procedural grouping was applied if the study investigated procedures commonly performed as part of laboratory routines, breeding procedures, or veterinary treatments including anaesthesia and analgesia provision. The biological classification was reserved for those studies that investigated grimace scores resulting from inherent biological variation such as between sexes and strains, or as a result of difficult to control environmental variables such as circadian rhythms. Based on our classification, 65% (*n* = 31) of studies used animal models, 31% (*n* = 15) looked at husbandry/procedural interventions, and 4% (*n* = 2) investigated biological variation in grimace scores. It was considered that the interventions applied would lead to pain arising of substantially different natures. We utilised a published pain classification system [73] for the assignment of studies based on pain type (Figure 3). Figure 4 presents a sub-classification of the type of animal models or procedures used in the included studies, with the expected pain type assumed to result. The animal model groupings are based on that presented by Hau and Shapiro, 2010 [74]. It should be noted that whilst some studies may have had a primary focus on evaluating response to one intervention, they may have reported on the impact of other factors, for example, sex differences. In reporting, we considered evidence from all studies irrespective of the classification assigned.

### 3.3. Mouse Characteristics

The included studies used a wide range of inbred strains and outbred stocks of mice. The C57BL/6 strain was used in the majority of studies (38% of uses), followed by the outbred ICR/CD-1 (24%). Transgenic or knockout/in strains of specific relevance to the research questions investigated in the publications were commonly used (14%). Figure 5 illustrates the relative uses of the various strains. Excluding the mutant, transgenic, and other categories, 45% of the mice used were black-coloured, 44% were white-coloured, and 11% were brown/agouti. Considering standard inbred or outbred strains/stocks only, eight studies used more than one strain [33,53,54,57,60,61,65,66]. Only three of these studies directly contrasted grimace scores between the strains [33,53,54]. The directions of effect for grimace scores in these comparisons are presented in Figure 6. There are some differences in strain effects on grimace scores between the sexes.

Male mice only were investigated in 40% (*n* = 19) of the studies, females in 21% (*n* = 10) of the studies, with 36% (*n* = 17) of the studies investigating both sexes (contrasted for sex differences in Table 2). Sex of mice was unreported in one study [41].

### 3.4. MGS Measurement Methods

The majority (88%) of studies evaluated MGS by retrospective scoring via photographs obtained directly via camera use, or extracted as stills from video footage, as reported in the original study [20]. To date, only five studies have used real-time methods [26,31,38,45,54] with three of these studies directly contrasting the results with those obtained from retrospective scoring [26,38,54]. One study did not state the method of MGS scoring [48]. The breakdown of collection method and timing is detailed in Figure 7.

The real-time methods used varied between the studies. In Miller and Leach, 2015, the mice were observed three times during a 10 min period and scored on each action unit as per the original method to arrive at three scores for the period [54]. Gallo et al., 2020, assigned a single score to each mouse after a 30 s observation period [38], and Bu et al., 2015 [31], assigned a single score on provoking abdominal pain in a pelvic pain model. The methods used in Hsi et al., 2020, are unclear but real-time methods appear to have been used [45]. However, Chartier et al., 2020, assigned a score every 15 s for 15 min and then calculated final scores based on averaging of scores across three 90 s periods to account for any effect of novelty of the box [26].

In the studies performing direct comparison, live scores were found to be significantly lower than corresponding retrospective scoring in two of the studies [26,54]. In the final study [38], a PCA produced a component where real-time MGS and image scoring were highly intercorrelated (with nesting behaviour as a third factor).

The original study described the MGS in terms of five FAUs. However, in 18 (38%) of the studies, scoring was modified by excluding specific action units, or in one case by combining the cheek and nose bulge action unit into one [50]. In the studies that used four action units for scoring, whisker position was the action unit excluded in the majority (60%) of cases (Figure 8). The method of combining the scores to arrive at a final score for the photograph or time point (real-time scoring) was in the majority of studies (36/48) by averaging of individual action unit scores (yielding a maximum score of 2). In 10 studies, summation of the individual action units scores was performed to arrive at the final score (maximum score of 10 for five FAUs). The method of achieving the final score was unclear in the remaining two studies [28,58]. A number of studies accounted for individual responses to pain by using mean difference scores in data presentation and analysis to correct for baseline grimace scores. For studies where the whisker position FAU was excluded, 50% of the studies used mice (6/12) that were black coloured, 33% (4/12) white, and 17% (2/12) brown coloured (*Χ^2^*(2, *n* = 12) = 3, *p* = 0.22).

A range of study durations were used in included studies, often with multiple time points being assessed within a single study. Duration of MGS assessment ranged from directly after the intervention to over a month following. This is illustrated in Figure 9, categorised by expected pain type. Refer to Table 1 for details regarding the interventions applied in the studies.

### 3.5. Corroborating Methods of Affective State Assessment Used

A range of alternate methods for assessing animal affective state were utilised in 37/47 (79%) of the included studies (Figure 10). These methods were largely behavioural in nature but did include measures of physiology, such as corticosterone analyses or bodyweight (being an expression of feeding behaviour). The most common measures used in rank order were: the use of von Frey filaments for the assessment of mechanical allodynia, bodyweight, and general clinical/disease scoring which may have been tailored to the model used, e.g., EAE scoring scheme, burrowing behaviour, pain-related behaviour scoring such as the use of composite pain measures, and open field tests for activity and locomotion. In the majority of cases (31 studies), data from these tests corroborated MGS scoring. In the remaining studies, either no association was seen with the chosen measures [26,53,58,72], or there was unclear reporting or a lack of direct comparison in the same animals [39,45].

### 3.6. Impact of External Factors on MGS

#### 3.6.1. Circadian Rhythm

In the majority of the studies, there was no specific reporting of light cycle stage for the recording of MGS data. It was assumed that given the lack of reporting, these were performed during the light stage. Five (11%) studies either reported conducting recording during the dark stage or timelines of measurement suggested that both stages would be crossed [27,47,51,54,60]. However, only three of these studies performed an examination of circadian rhythm effects [51,54,60] (studies and their impact on the MGS are reported in Table 3).

#### 3.6.2. Variability Arising from Observers

A number of studies (20/48) utilised more than one observer for ascertaining grimace scores. Ten of these studies (Table 4) specifically reported the metrics associated with agreement between the observers, that allowed them to combine the results with an assurance of external reliability.

## 4. Discussion

In this paper, we present the first comprehensive overview of all studies investigating the MGS, assimilating information on the types of animal models/conditions where the MGS has been applied, methods applied, and external factors affecting the validity of the technique. It is intended that this assimilation will guide the future validation and use of the MGS by researchers, and thus promote wider scale implementation of the method. The key findings of our assimilation are discussed below.

### 4.1. Methods Used

To date, the majority (88%) of uses of the MGS in biomedical research settings have used retrospective recording through collection of video footage, and subsequent still extraction, or primary collection of photographic images. Retrospective scoring brings some key advantages when using the MGS as a research outcome measure. These methods provide a greater degree of certainty in the findings by allowing for the possibility of re-confirming scores and thus replicating the data, utilising multiple observers for cross-checking, and allowing scoring to occur at a time that suits the researcher [6]. This can all occur without the potential modulating influence on the scores of a human observer [65]. Whilst, not discussed in the included studies, an assumed challenge in using cameras to secure facial images is the need to achieve a face-on shot. This might be achieved by using a ‘burst’ mode to take photos in rapid succession, or by manual performance by an observer. However, this does raise concerns about the effect of observer presence on grimace scores and the impact of any noise produced by the camera when photographs are taken.

A real-time method has advantages for clinical pain assessment, since scores can be attained quickly, to allow immediate action such as applying a humane endpoint or providing analgesics. The method may also provide some advantages in a research scenario by limiting the need for post-processing of images, which is invariably time consuming [16]. To date there has been limited evaluation of real-time scoring in mice, and of the five studies that have utilised this, only three directly contrasted this with validated retrospective scoring methods. Two studies found live scores to be lower than corresponding retrospective scoring [26,54]. A reason proposed for the lower scores resulting from live scoring is that the nature of the face changes rapidly during live scoring, whereas in images, for example, random selection will lead to the capture of blinking which is assigned a high score, contributing to relatively higher scores [54]. Alternately, as proposed by Chartier et al., 2020 [26], the presence of a human observer in real-time scoring may influence mouse performance of the facial action units; increased alertness could lower the grimace scores through eye widening and ‘pricking’ of the ears. It should be noted that there are considerable differences in the technique used for collection of real time data with some studies basing a score on a single observation point [31,38], as opposed to mathematical integration of several scores taken across a period [26,54]. The former would be simpler in a clinical context but may be associated with loss of sensitivity and validity. In spite of this, point grimace scores were determined to move in the expected direction of effect in these studies, implying validity. In rats, there has been dedicated study into methods of real-time scoring and their relationship with retrospective scoring [75,76], and this is clearly needed in mice.

Whilst 62% of studies did use all of the five original described action units for scoring, in a significant proportion of studies (37%), scoring was modified by excluding specific action units, or combining units. Most of these adaptations involved excluding whisker scoring, which seems to be regarded as hard to visualise/score [16]. It has been suggested by some authors that this difficulty in scoring whiskers is related to black coat colour [33,50]. However, this proposition is not supported by our synthesis, which implies that whiskers are excluded from scoring at similar rates independent of coat colour (although study numbers are low). There may also be an impact of inexperience in scoring on the ability to accurately identify action units; for instance, Hohlbaum et al., 2020 [44], demonstrated that cheek and nose bulge scoring had reduced inter-observer agreement compared with orbital tightening, with inexperienced scorers having even reduced accuracy.

### 4.2. Validity of the MGS across a Range of Pain Types

The MGS is described as a measurement of pain, i.e., it has face validity for pain. There is clear evidence from the included studies that the MGS changes in response to painful events and is modified by analgesics, further supporting this proposition—see, e.g., [37,49,51]. However, another important aspect of the validity of a pain measure is the extent to which the technique measures pain, and is not influenced by other conditions such as sickness behaviour—in other words, whether it has construct validity. This review assists in evaluating these concepts in a number of ways.

It is clear that whilst the majority of the studies examining the MGS are conducted by researchers in the pain field, there has now been use of the technique across a range of non-pain focussed animal models. The technique being especially utilised in the oral health science and neuroscience fields. There has also been significant focus on the technique in husbandry and welfare investigations in mice, with a focus on the effects of surgery and analgesic administration on the score, and by inference pain. In the majority of these models, especially over an acute timeframe, the MGS has good utility. However, even though use of the technique has increased over the past decade, 48 studies is a small fraction of all the studies being conducted in laboratory mice. It is surprising that more researchers have not taken the opportunity to include the technique in their study. This may be due to a lack of awareness by researchers outside the pain and veterinary research fields of both the technique and its validity. It is hoped that this review will promote awareness among these researchers, but there is probably a significant role for animal ethics committees in this dissemination effort.

In the original study by Langford et al., 2010, it was considered that the MGS was only suitable for measuring acute pain, based on the lack of grimace response when models of chronic pain were applied [20]. This would make sense from an evolutionary perspective since, as prey animals, mice may learn to control a facial pain response to avoid predation [51]. However, later studies question this assumption. Figure 8 provides clear evidence that across a range of different expected pain types, grimace scores are detected up to a month post-initial insult in situations where pain might be expected. This evidence is particularly strong for neuropathic pain, which might be expected to be longer lasting and has been investigated in a reasonable number of studies. In visceral or mixed pain, the MGS also appears to be able to detect an effect, but there have been limited studies, and it should be noted that the studies into mixed pain both come from the same laboratory looking at pain in one model of breast carcinoma [34,35]. There is clearly a need for future study in models where these types of pain are expected. To date, no studies have shown the existence of changed grimace scores at timepoints greater than a month after the assumed painful treatment. However, only two studies specifically looked at these timepoints, and there is the possibility that pain was not actually present at these times, especially in one of the studies, which utilised a relapsing-remitting colitis model induced by DSS [26]. Interpreting findings at these later timepoints is made more challenging given the lack of other validated measures of pain against which it is possible to corroborate MGS findings.

A range of physiological and behavioural outputs were measured in the included studies, which lend support to the proposition that the MGS has good construct validity. These included the use of assessment of mechanical allodynia, general clinical scoring, pain behaviour scoring, or indicators of luxury behaviour, such as nest building or burrowing. In the main, outcomes from these tests moved in the same direction as mouse grimace scores, suggesting convergent validity. However, out of all the measures assessed, arguably only a couple are specific to pain and are plagued by the same issue that surrounds MGS validation; that of establishing incontrovertibly what they are measuring. For example, burrowing and nest building behaviour are largely taken to be generalised indicators of well-being or affective state [77], and are modified not just in response to pain, but sickness behaviours (see, e.g., [78,79,80,81]). Composite pain behaviour scoring and use of von Frey testing are specific to pain and are therefore more reliable corroborating measures. However, the debate around the differences between nociception and pain needs to be borne in mind (see [82] for full discussion). The former is a physiological function, but a reaction to a stimulus does not necessarily signify the experience of pain. Therefore, the widespread historical use of stimulus-evoked tests, such as the von Frey filaments, may be a contributing factor to the poor translation rates in pain research [82]. One of the key cited advantages of the MGS is that it measures spontaneous pain [10]. Based on this discussion, perhaps the most reliable corroborating measure against which to assess the MGS is another readout of spontaneous pain, with composite pain behaviour scoring being the only measure to completely fulfil this description. In the studies that compared these two readouts, the direction of effect was aligned, but the studies are few in number (5) [28,47,49,53,56].

Another finding of this review that questions the construct validity of the MGS is the change in grimace scores in response to techniques that would not be expected to elicit pain. Out of the eleven studies that examined the MGS over the 24 h period after an intervention that was expected to elicit no or momentary pain, six found grimace score elevations. A further examination of these studies shows that three of the studies were examining the effect of anaesthesia/analgesia on grimace scores [42,43,53]. In general, both inhalational [42,53] and injectable anaesthetics [43] increase scores, in the absence of a presumed painful event. However, whilst analgesia might similarly be expected to elevate scores, in two studies, both tramadol [46], and buprenorphine [53] were not determined to have any impact. The impact of the anaesthetics is short-lived, having resolved by 24 h. It is postulated that this could be related to a ‘hangover’ or sedative effect remaining after the procedure, which could be envisaged to lead to eye closure as in sleep. However, perhaps a lingering muscle relaxant effect could similarly affect the other action units. The evidence on an elevation with inhalational anaesthetics is also not clear, with a strain effect being identified in the Miller et al., 2015 [53] study. The study by Sorge et al. [65] is mechanistically different to the other studies within this group, since exposure to a painful insult was applied, with differences in grimace response shown to result from a form of male pheromone-induced stress analgesia. The remaining two studies found increased grimace scores as a result of blood sampling [52] and handling and identification [64]. In the former [52], facial vein and retrobulbar bleeding increased scores in the immediate post-procedural period. This study also provides further evidence for the effects of isoflurane on the MGS with increased scores seen in anaesthetised compared to sham handled groups. In the study of [64], increased scores were seen as a result of tail handling and ear tagging. There are several points of relevance here in relation to MGS construct validity. Firstly, the blood sampling interventions applied are likely to produce momentary pain as opposed to no pain [83], so evidence of a change actually supports construct validity. Secondly, tail handling has been suggested to be aversive rather than painful [84], so an effect does call into question the specificity of the scale for pain (although it is worth noting that a previous study found no effect of handling [57]). Thirdly, whilst blood sampling only caused immediate post-procedural changes in MGS (later time points were not examined), differences between groups for handling and identification often persisted for 24 h, when it might be assumed that any pain would have resolved, even, as demonstrated in this study, at a time point when inflammation remained [64]. Interestingly, there was also non-convergence of findings relating to inflammatory response and MGS with tunnel handled mice demonstrating a greater response than tail-handled animals.

### 4.3. Reliability

Pain scales should be reliable—that is, they should produce similar results whenever they are used [85]. This requires that between animal, intra-animal, and temporal variations are minimised unless they result from differences in pain experience. Reliability impacts validity, since if errors in measurement are significant, the scale no longer performs well at assessing pain [16]. The included papers assessed a number of measures of reliability including within observer variability (intra-observer), between observer (inter- observer), and across site variability.

Whilst a fair proportion of the studies investigating grimace scales utilised more than one observer for scoring, only 50% of these analysed and reported on between observer metrics. This represents a significant loss of data on the reliability of these scales. This raises the question of whether these data were not analysed, or not reported, perhaps because of low agreement. If there was more ability and uptake of protocol registration in pre-clinical studies, this question may not have arisen. Moreover, in encouraging the use of these scales for practical welfare assessment as clinical tools, this question is important; few institutions will be able to rely on the same, single observer to perform all scoring.

Based on the limited evidence available, inter-rater agreement generally ranges from good to excellent. However, a recent study [44] did suggest that this may change over time, with differences potentially being obscured by the assimilation of all data. This is a factor that should be considered in future studies. Related to this, there may also be differences in scores for similar treatments when taken across laboratories [46]. It is not clear whether this relates to inter-observer differences, or differences in housing/test conditions, but does call into question the external validity of MGS results [46]. However, importantly, this study did find that whilst values across research centres were numerically different, the direction of effect was similar, so general validity was maintained. Fewer studies have reported on intra-rater variability, although the study by Mittal et al. 2016, which used a large number of coders (6), did report significant within coder variability in three individuals [59]. All of these findings raise the question of whether training and experience in the use of the scales impacts on reliability. Few studies have specifically examined this, and detailed information on training was rarely provided in the included studies. Evidence for a training effect is currently conflicting, with one study [44] suggesting greater consistency if scorers were experienced, whilst another study [64] finding good correlation between novice and expert scorers. The impact of and type/frequency of training needed to produce reliable grimace scores is an area that needs further research, especially if the technique is going to gain more widespread acceptance as a pain assessment tool. This is also a particular consideration for real-time scoring, which needs to be performed quickly and does not offer the opportunity for re-review of collected images.

### 4.4. The Impact of Biological Variation and the External Environment on the MGS

The synthesis demonstrates that there are a number of features of biology and the external environment that influence grimace scores. These include the influences of strain, sex, the circadian cycle, and observers. These differences should be considered in future investigations of grimace scores, especially in the development of intervention scores.

A limited number of studies have directly contrasted more than one strain [33,53,54]. It is difficult to draw any conclusions on the impact of strain on grimace scores, since there appears, at least on the basis of one study [54], to be interactions between sex and strain on grimace scores. In general, with some exceptions due to sex differences, it appears that the strain order from propensity to score low to high is C57BL/6, CD1, C3H/He, and BALB/c. However, it is worth noting that much of this information on strain differences comes from one study [54], where a painful insult was not applied. This may be of relevance, particularly on consideration of the interaction between sex and strain, since it is well established that there are differences in pain thresholds between male and female rodents, with females having a lower pain threshold in response to a variety of nociceptive inputs [86]. It is interesting to note that this strain ranking shows no obvious trend based on coat colouration, implying that inability to score individual action units due to this may have minimal impact on scores obtained.

Evidence as to the presence or nature of any differences in scores as a result of sex is far from settled. The majority of studies that compared sex differences within the same strain found no differences in scores. In regard to the minority of studies that did find sex differences, there is a fairly even split between those that found that scores were lower in females and vice versa. This is perhaps surprising given the finding using traditional pain assays that female rodents have a lower pain threshold in the face of hot thermal [87], chemical [88], inflammatory [89], and mechanical nociceptive insults [90]. However, varied findings in relation to sex differences are not uncommon in these other models, and probably arise due to differences in study design as well as genotype [91]. The absence of a sex effect in the majority of the studies that evaluated both sexes may also speak to a lack of sensitivity of the scoring, whereby differences are present, but cannot be discriminated. Another more general finding arising from the assimilation is that in spite of increased promotion of the use of both sexes in preclinical research due to concerns about translation [92,93], the majority of studies used one sex (predominantly males). Even when two sexes were used in the included studies, an opportunity was often missed by failing to make direct comparisons between them.

Circadian rhythms commonly apply to biological and physiological processes in animals [94]. Mice, as nocturnal animals, are active mainly during the dark phase [95]. The strength of this circadian clock is such that even in constant darkness, this pattern of activity will persist, despite the absence of external cues [95]. There is also evidence of a circadian rhythm in pain sensitivity across a range of animal species [96,97], potentially brought about by a rhythm associated with opioid peptide production [98,99]. Considering that general levels of activity are likely to confound behavioural measurements particularly (although not exclusively), it follows that experimental protocols would control for this, and report on time of testing. This also raises the question of whether performing behavioural tests in the light phase is a major methodological error [100]. Given this, it is surprising that many of the included studies failed to report on the timing of MGS measurements; this being an item in the updated ARRIVE guidelines recommended set [101]. Given the lack of dedicated study and reporting deficiencies, there is limited evidence to support or refute an effect of circadian rhythm on the MGS. However, two studies hint at potential differences [51,60] with a suggestion of higher scores or pain in the dark phase. Nevertheless, Rea et al, 2018 did discuss that light transition appeared to cause decreases in orbital tightening and nose bulge, and it is not clear whether this effect would have persisted once acclimatised to the light [60].

Observer effects on the scale have rarely been investigated. This is unsurprising given that the majority of studies using the MGS have utilised retrospective analysis for scoring. However, as previously discussed, observer effects may be relevant when photography is used, and are of clear importance in real time scoring since it is well established from animal behaviour research that a human observer may influence animal behaviour [102]. There is some suggestion from other species of minimal impacts on grimace scores by human observers (see, e.g., [75]). However, this needs dedicated investigation in the context of mice. Furthermore, the nature of the observer may be important in determining their impact on scores. For example, Sorge et al., 2014, demonstrated that the presence of human males led to a stress-induced analgesia and reduced grimace scores, and familiarity with the observer may also be a factor in response [16].

### 4.5. Conclusions and Recommendations for Future Research

This review has assimilated all primary literature to date on the MGS. It is concluded that the MGS has utility across a range of animal models and expected pain types. There do, however, appear to be some differences arising as a result of biological variation, such as sex or strain of mouse. These variables need consideration in study design or analysis to account for them appropriately. There is also some limited evidence that the MGS may not be wholly specific to pain. However, this evidence mainly comes from studies into husbandry or drug interventions, the latter generally only having a short-term effect, which can likely be explained by the pharmacological effects. It would be interesting to delve further into any potentially non-pain-related grimace effects in animal models where other symptoms might be assumed to co-occur with pain—for example, sickness behaviour. This could potentially be achieved by using analgesics to eliminate the pain response, although of course the risk of drug confounding would need to be considered.

Further research is needed on the use of the MGS as a real time method, and how this can be done to maintain validity of the method, whilst being practically feasible. Related to this is the question of how reliable scoring between observers is, and what type of training (if any) is needed to maximise between observer agreement. Finally, whilst there is suggestion from studies in this synthesis [65], and others [103], that there is a social modulation of pain by conspecifics and the presence of other species, there has been little investigation of this fascinating area in the context of grimace responses.

## Figures and Tables

**Figure 1 animals-11-00673-f001:**
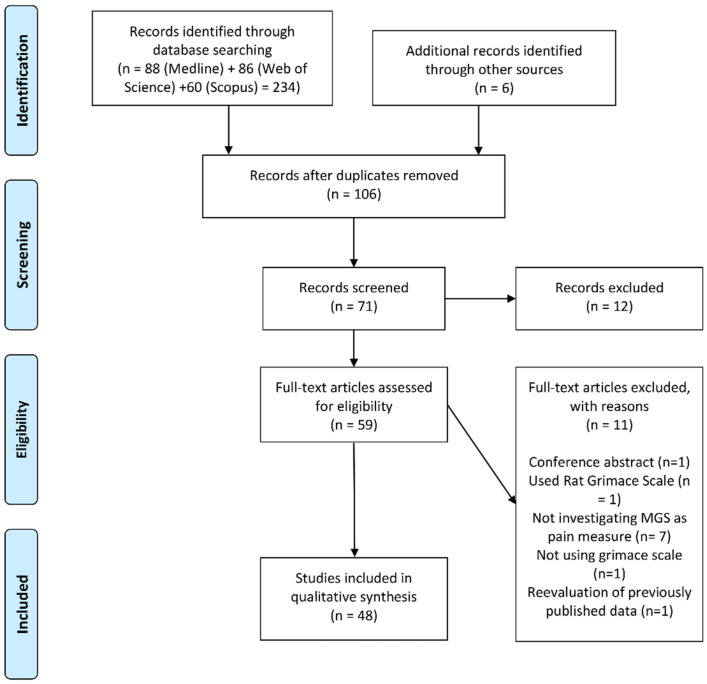
The PRISMA [29] flow diagram for the review detailing the database searches, the number of abstracts screened, and the full texts retrieved.

**Figure 2 animals-11-00673-f002:**
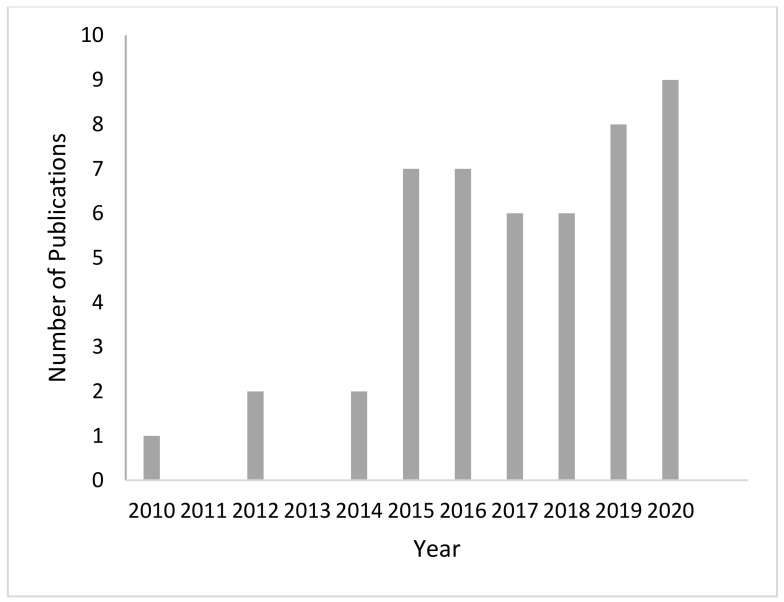
Number of publications per year investigating the Mouse Grimace Scale (MGS) in relation to pain.

**Figure 3 animals-11-00673-f003:**
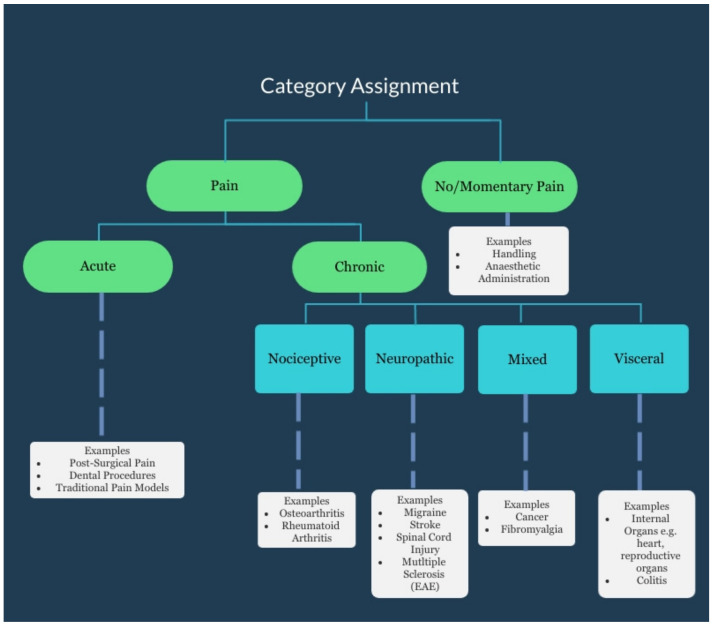
Pain classifications used to guide assignment of studies to categories. Adapted with permission from Springer Nature and Copyright Clearance Center: Springer Nature, Nature Reviews Drug Discovery, Pain Market, Melnikova, COPYRIGHT 2010 [73]. For specific category assignment for included studies, refer to Table 1.

**Figure 4 animals-11-00673-f004:**
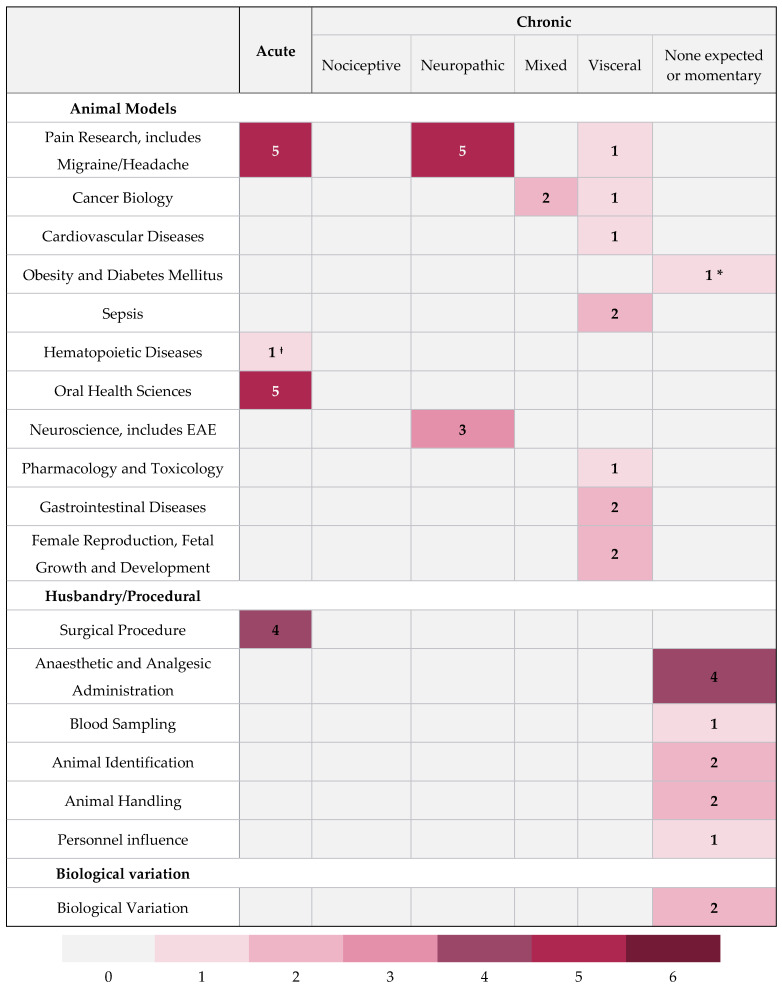
Heat map contrasting interventions used with the type of pain expected to be elicited. Colouration/number in box represents number of studies. Whilst some studies could arguably have been included in multiple categories to simplify reporting, one category has been assigned. ^ǂ^ Pain in this study, Mittal et al., 2016 [59], was assigned as acute since it was induced by cold stress, although sickle cell pain can be neuropathic in origin. * Model of Hsi et al., 2020 [45], did not relate to a neuropathy.

**Figure 5 animals-11-00673-f005:**
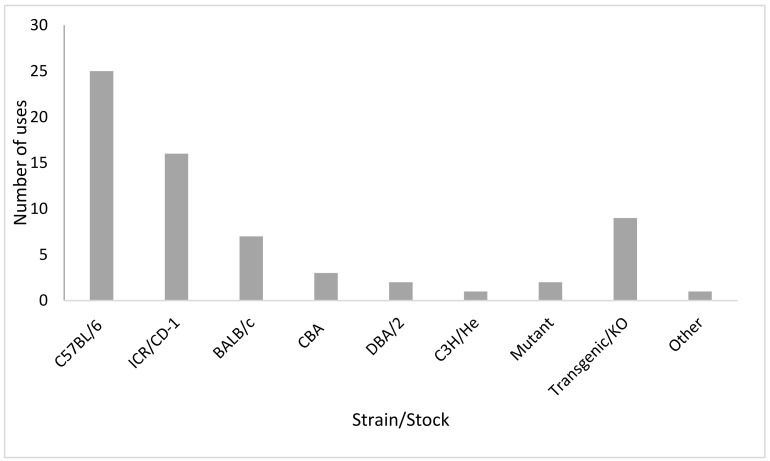
Representation of the different mouse strains and stocks used in the published papers. Note that a number of papers used more than one strain. The ICR and CD-1 nomenclature has been considered to represent the same stock. Other includes hybrid or recombinant strains.

**Figure 6 animals-11-00673-f006:**
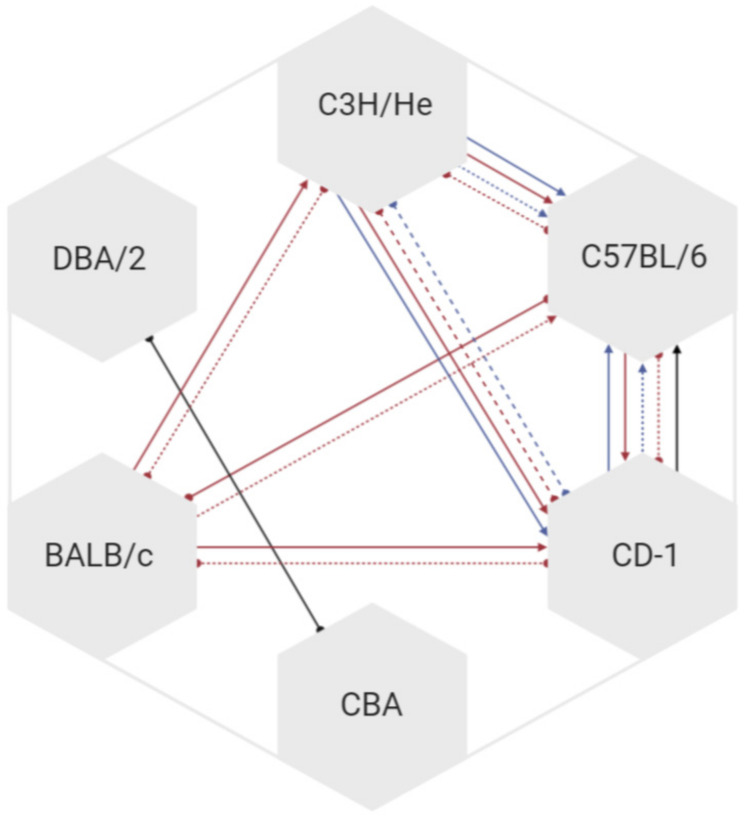
Network map comparing MGS scores between strains. Each line represents a study effect. The direction of the arrow represents that the strain at the arrowhead responded with a lower MGS score. Red lines indicate a comparison between female mice, blue lines indicate comparison between male mice, and black lines indicate comparisons where sex was not separated. A solid line indicates that a live score was used, a dashed line indicates that a retrospective score was used.

**Figure 7 animals-11-00673-f007:**
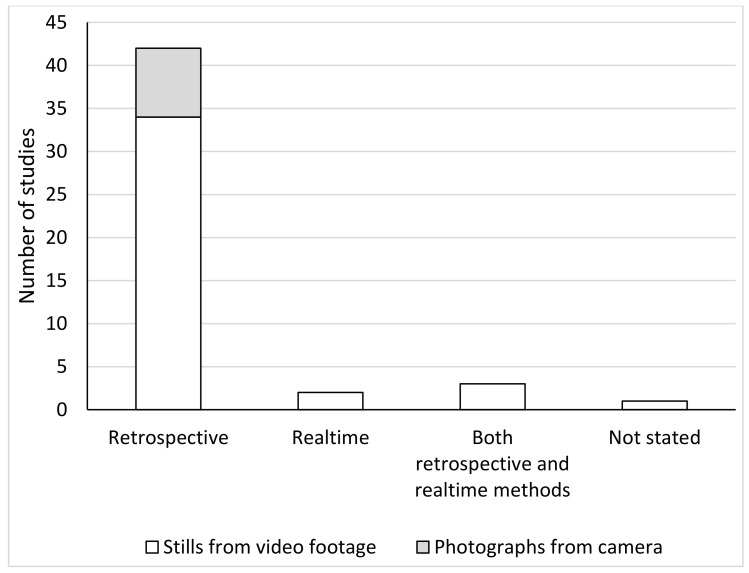
Collection methods used in included studies.

**Figure 8 animals-11-00673-f008:**
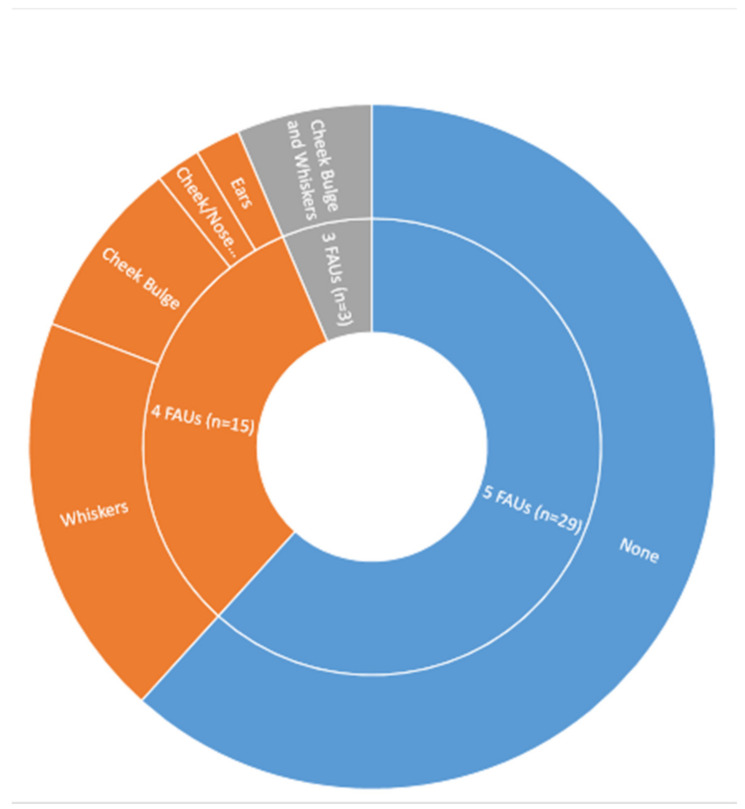
Facial action units (FAUs) utilised for scoring in the included studies. n represents study number. Specific action units were generally excluded as described here, although in one study two of the action units were combined.

**Figure 9 animals-11-00673-f009:**
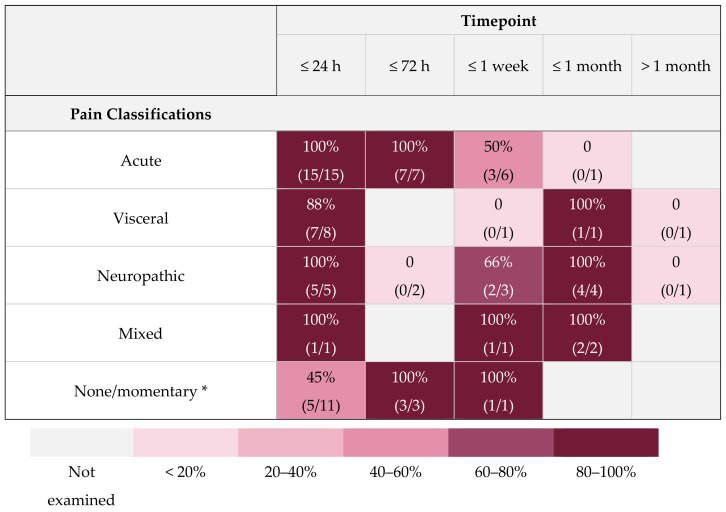
Heat map contrasting type of pain expected to arise from the interventions with the time points after the intervention investigated. Colouration gradation represents percentage of studies where grimace scores moved in the expected direction of effect, with increased shading indicating a greater number of investigations—for example, 100% of studies evaluating procedures likely to cause acute pain showed increased MGS scores within the 24 h after the intervention. * Consider that no change in MGS score is expected.

**Figure 10 animals-11-00673-f010:**
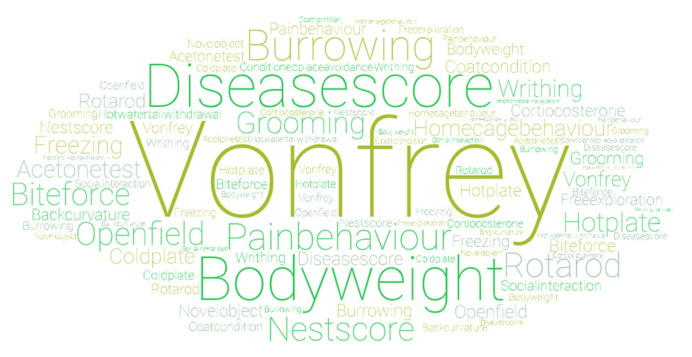
Word cloud illustrating corroborating methods of affective state assessment used in the included studies. Size of the word illustrates their relative frequency of use.

**Table 1 animals-11-00673-t001:** List of included studies.

Reference	Study Design ^§^	Strain/Stock	Age/Weight	Sex	Type of Intervention	Intervention (Model Created or Procedure Investigated)	Pain Classification Assigned	Intervention Effect on MGS Score *
Akintola et al., 2017 [30]	RCT	C57BL/6	10–12 weeks	M	Animal Model	Chronic constriction injury model for pain	Neuropathic	↑
Bu et al., 2015 [31]	RCT	BALB/c	6–8 weeks	F	Animal Model	Chronic pelvic pain	Visceral	↑
Burgos-Vega et al., 2019 [32]	RCT	ICR	6–8 weeks	M,F	Animal Model	Migraine	Neuropathic	↑
Chartier et al., 2020 [26]	RCT	C57BL/ 6JArc	8 weeks	F	Animal Model	Colitis-associated colorectal cancer	Visceral	Nil
Cho et al., 2019 [33]	RCT	CD-1, C57BL/6N	7–9 weeks	M,F	Animal Model	Craniotomy with different analgesics	Neuropathic	↑ (reduced by analgesics)
de Almeida et al., 2019 [34]	Pre-test, post-test	BALB/c	20–30 g	F	Animal Model	Cancer-induced nociception	Mixed	↑
de Almeida et al., 2020 [35]	RCT	BALB/c		F	Animal Model	Cancer-induced nociception	Mixed	↑
Duffy et al., 2016 [36]	RCT	C57BL/6J	10–12 weeks	F	Animal Model	Experimental autoimmune encephalomyelitis (EAE)	Neuropathic	↑
Dwivedi et al., 2016 [27]	RCT	Transgenic (BL/6 background) PCSK9 KO mice and PCSK9 overexpression	10–12 weeks	M,F	Animal Model	Caecal Ligation and Puncture model of sepsis	Visceral	↑
Faller et al., 2015 [37]	RCT	C57BL/6J, transgenic overexpressing creatine transporter in the heart (BL/6 background)	12–16 weeks	F	Animal Model	Myocardial infarction created through thoracotomy	Visceral	↑ (reduced by analgesics)
Gallo et al., 2020 [38]	RCT (factorial)	Crl:CD1(ICR)	8–9 weeks	M	Husbandry/Procedural	Carotid artery catheterisation	Acute	↑
Guo et al., 2019 [39]	RCT	C57BL/6	9 weeks	M,F	Animal Model	Orofacial pain	Acute	
Hassan et al., 2017 [28]	RCT	C57BL/6N, PYY knockout	10 weeks	M	Animal Model	Colonic nociception	Visceral	↑
Hassler et al., 2019 [40]	RCT	ICR, C57BL/6J, and PAR2 (BL/6 background)	20–30 g	M	Animal Model	Migraine	Neuropathic	↑
Herrera et al., 2018 [41]	RCT	CD-1	18–20 g	nr	Animal Model	Bothops Asper venom	Visceral	↑
Hohlbaum et al., 2017 [42]	RCT	C57BL/6JRj	11–13 weeks	M, F	Husbandry/Procedural	Isoflurane anaesthesia	None/momentary	↑ (female mice only)
Hohlbaum et al., 2018 [43]	RCT	C57BL/6JRj	11–13 weeks	M, F	Husbandry/Procedural	Ketamine/xylazine anaesthesia	None/momentary	↑
Hohlbaum et al., 2020 [44] ^⁋^	Quasi-experimental	C57BL/6JRj	11–13 weeks	M, F	Biological	Ketamine /xylazine anaesthesia	N/A	Study investigated inter-observer reliability in scoring
Hsi et al., 2020 [45]	RCT	C57BL/6N	7–9 weeks	F	Animal Model	Animals with hypoglycaemia following Roux-en-Y Gastric Bypass surgery	None/momentary (outcome of interest is the hypoglycaemia)	Nil ^ǂ^
Jirkof et al., 2020 [46]	RCT	C57BL/6J		F	Husbandry/Procedural	Tramadol treatment effect on MGS between laboratories	None/momentary	Nil
Jurik et al., 2014 [47]	RCT	TRPV1 knock-out (BL/6 background)	8–16 weeks	M	Animal Model	Abdominal constriction test and acute pancreatitis as models of pain. Effects of knockout versus wildtype genotype	Visceral	No effect of genotype on MGS scores
Kim et al., 2015 [48]	RCT	ICR		M,F	Animal Model	Hyperalgesic priming via IL-6 and Carrageenan injection	Acute	↑
Langford et al., 2010 [20]	RCT	CD-1 (ICR:Crl)	6–18 weeks	M,F	Animal Model	14 models of pain	Acute	↑
Leach et al., 2012 [49]	RCT	CD-1		M	Husbandry/Procedural	Vasectomy surgery	Acute	↑ (reduced by analgesics)
Mai et al., 2018 [50]	RCT	C57BL/6J	8–12 weeks	M	Animal Model	Caecal ligation and puncture model of sepsis	Visceral	↑
Matsumiya et al., 2012 [51]	RCT	CD1 (ICR:Crl)	6–8 weeks	M,F	Husbandry/Procedural	Ventral ovariectomy and response to analgesics	Acute	↑(reduced by analgesics)
Meyer et al., 2020 [52]	RCT	C57BL/6J	10–12 weeks	M	Husbandry/Procedural	Common recovery blood sampling routes (facial vein, retrobulbar, tail vein with anaesthetic and handling control)	None/momentary	↑—anaesthetic, facial vein bleeding, or retrobulbar compared to handling
Miller et al., 2015 [53]	Pre-test, post-test	CBA, DBA/2		M	Husbandry/Procedural	Isoflurane anaesthesia and buprenorphine analgesic	None/momentary	Nil (↑ by isoflurane in DBA/2 strain)
Miller and Leach, 2015a [54]	RCT	C57BL/6, C3H/He, CD-1 BALB/c	8 weeks	M,F	Biological	Impact of sex, strain, time of day or habituation	None/momentary	Nil-order ↑- males compared to females (but strain dependant and not always consistent) Strain effects present Time of day effects with sex and strain differences
Miller and Leach, 2015b [55]	RCT	C57BL/6	8 weeks	M	Husbandry/Procedural	Ear notching and analgesic effects	None/momentary	Nil
Miller et al., 2016 [56]	Pre-test, post-test	CBA	25.6–28.7 g	M	Husbandry/Procedural	Vasectomy surgery	Acute	↑
Miller and Leach, 2016 [57]	RCT	CBA, DBA/2		M	Husbandry/Procedural	Handling method: tail versus tube	None/momentary	Nil
Mitchell et al., 2020 [58]	RCT	ArcCrl:CD	12 weeks	F	Animal Model	TNBS-induced Crohn’s-like colitis	Visceral	↑
Mittal et al., 2016 [59]	RCT	Transgenic HbSS-BERK (with relevant controls)		M,F	Animal Model	Sickle cell disease and effects of cold	Acute	↑—in females, cold also had impact
Rea et al., 2018 [60]	RCT	C57BL/6J, CD1	10–14 weeks	M,F	Animal Model	Pain as result of migraine	Neuropathic	↑
Rosen et al., 2017 [61]	RCT	CD-1 (Crl:ICR), Nude (Crl:CD1- Foxn1nu), C57BL/6J, C57BL/6-Rag1 ^tm1Mom^/J, mutant mice lacking expression of the Oprd1 (-opioid receptor) gene	7–12 weeks	M,F	Animal Model	Pregnancy analgesia after inflammatory insult induced by administration of complete Freund’s adjuvant (CFA)	Visceral (pregnancy state).	↑—in late-pregnant mice compared to nulliparous females
Rossi et al., 2020 [62]	RCT	Mixed CD-1 and C57BL/6J background	17–21 weeks	M,F	Animal Model	Tooth pulp injury	Acute	↑
Roughan et al., 2016 [63]	RCT	BALB/c	25–30 g	M	Husbandry/Procedural	Handling method: tail versus cupping at time of surgery	None/momentary	Nil (although surgery itself increased MGS)
Roughan and Sevenoaks, 2019 [64]	RCT	BALB/cAnNCrl	10–13 weeks	M,F	Husbandry/Procedural	Ear tattooing and tagging, with tail handling method or tunnel	None/momentary	↑ tail versus tunnel ↑ males versus females ↑ ear tagging versus tattoo
Sorge et al., 2014 [65]	RCT	CD-1 (ICR:Crl), C57BL/6J	6–12 weeks	M,F	Husbandry/Procedural	Effect of gender/gender-specific and other animal pheromones on response to nociceptive assays	None/momentary (intervention of interest is the pheromones)	↓ with male observer or male’s T-shirt compared to no observer
Serizawa et al., 2019 [18]	RCT	C57BL/6J	7 weeks	F	Animal Model	Experimental autoimmune encephalomyelitis (EAE)	Neuropathic	↑
Tillu et al., 2015 [66]	RCT	ICR, C57BL/6	20–25 g	M	Animal Model	Hyperalgesic priming	Acute	↑
Tuttle et al., 2018 [67]	RCT	CD-1 (ICR:Crl)	6–12 weeks	M,F	Husbandry/Procedural	Ventral ovariectomy and response to analgesics, xymogen assay (validation of automated scoring)	Acute	↑ (reduced by analgesic)
Wang et al., 2017 [68]	RCT	C57BL/6, TRPV1 KO	8–12 weeks	M	Animal Model	Masseter inflammation	Acute	↑
Wang et al., 2018 [69]	RCT	C57BL/6, TRPV1 KO (C57BL/6 background), TRPA1 KO (mixed B6; 129 background)	8–12 weeks	M	Animal Model	Masseter inflammation	Acute	↑
Wang et al., 2019 [70]	RCT	C57BL/6	12 weeks	M	Animal Model	Orthodontic tooth movement	Acute	↑
Wu et al., 2016 [71]	RCT	C57BL/6	8–19 weeks	M	Animal Model	Spinal cord injury	Neuropathic	↑
Zhu et al., 2017 [72]	RCT	BALB/c	25–30 g	M	Animal Model	Orthodontic tooth movement	Acute	↑

^§^ The terminology randomised control trial (RCT) has been used to indicate use of a comparator with a parallel arrangement of study groups; however, randomisation was not necessarily performed at all or to a high standard in all studies. This was not specifically investigated as part of this review. ***** General consistent direction of effect ^ǂ^ no sham control so effect of model unknown ^⁋^ Study used photos obtained from Hohlbaum 2017 [42] study. In reporting, we have not considered this to represent an additional animal study for presentation of animal-focussed data.

**Table 2 animals-11-00673-t002:** Comparison of MGS scores between sexes.

Reference	Intervention	Strain	Direction of Effect (F vs. M)
[32]	Migraine	ICR	Not directly compared
[33]	Craniotomy with different analgesics	CD-1 C57BL/6N	= =
[27]	Caecal Ligation and Puncture model of sepsis	Transgenic (BL/6 background) PCSK9 KO mice and PCSK9 overexpression	Not reported
[39]	Orofacial pain	C57BL/6	Not reported
[42]	Isoflurane anaesthesia	C57BL/6JRj	=
[43]	Repeated ketamine anaesthesia	C57BL/6JRj	=
[48]	Hyperalgesic priming via IL-6 and Carrageenan injection	ICR	Not reported
[20]	14 models of pain	CD-1 (ICR:Crl)	=
[51]	Ventral ovariectomy and response to analgesics	CD1 (ICR:Crl)	=
[54]	Biological		**Live Scoring**
		C57BL/6	=
		C3H/He	F < M
		CD-1	F < M (at 1 time point)
			**Retrospective Scoring**
		C57BL/6	F > M
		C3H/He	=
		CD-1	=
[59]	Sickle cell disease and effects of cold	Transgenic HbSS-BERK (with relevant controls)	F > M
[60]	CGRP- induced migraine	C57BL/6J CD1	=
[61]	CD-1 (Crl:ICR), Nude (Crl:CD1- Foxn1 nu), C57BL/6J, C57BL/6-Rag1 tm1Mom, mutant mice lacking expression of the Oprd1 (-opioid receptor) gene	Pregnancy analgesia	Not directly compared for grimace outcome
[62]	Tooth pulp injury	Mixed CD1 and C57BL6/J background	=
[64]	Ear tattooing and tagging, with tail handling method or tunnel	BALB/cAnNCrl	F < M
[65]	Effect of gender/gender-specific and other animal pheromones on response to nociceptive assays	CD-1 (ICR:Crl) C57BL/6J	F > M (baseline values) Females displayed greater ‘male observer’ effect, e.g., increased reduction in grimace scores.
[67]	Ventral ovariectomy and response to analgesics, xymogen assay (validation of automated scoring)	CD-1 (ICR:Crl)	=

**Table 3 animals-11-00673-t003:** Studies that utilised grimace scoring in the dark stage of the circadian cycle and the impact on scores were reported.

Reference.	Reporting Detail	Intervention	Compared with Measures in Light (Y/N)	Direction of Effect for Comparison between Light and Dark
[51]	Conducted circadian study comparing light and dark recordings	Ventral ovariectomy and response to analgesics	Y	Compared mice which had surgery in the morning versus the evening with measurement timepoints of baseline, and every 6 h past surgery for 48 h There was no circadian effect on baseline MGS scores. However, mice operated on in the morning displayed larger MGS increases 12 h after surgery compared to 24 h, whilst mice operated on in the evening showed smaller increases at these time points. This suggests that mice experience higher levels of postoperative pain at night (dark phase)
[54]	No dark cycle recording but did compare MGS across the light phase	Biological	N	**Live scoring** There was no difference in MGS score between three time points (9 am, 12.30 pm, 4 pm) for C57BL/6, CD-1 or C3H/He mice. BALB/c mice showed a greater score at Noon compared to AM. **Retrospective Scoring** There was no significant difference in MGS scores between the three time points for CD-1, C3H/He or BALB/c mice. C57BL/6 mice showed a greater MGS scores at both Noon and PM time points compared to the AM time point
[60]	Performed a restrained grimace technique in dark (manipulated dark condition- not part of cycle)	CGRP-induced migraine	Y (compared bright light)	Grimace scores were higher in the dark than in bright light for the CD1 mice. Light transition led to decreased orbital tightening and nose bulge. C57BL/6J mice showed no significant difference between the CGRP-induced grimace in light and dark. Responses to CGRP were generally similar in direction as those recorded in the light.

**Table 4 animals-11-00673-t004:** Consistency metrics reported in the included studies. Inter-observer and inter-laboratory analyses were reported.

Reference	Number of Observers	Consistency Metrics
**Inter-Observer Variability**		
[37]	2	There was an excellent correlation between the two observers for MGS measurement (r = 0.98) assessed using Type II regression analysis. However, Bland–Altman analysis showed that the slope differed from unity with a bias towards higher MGS scores in one observer.
[44]	4 (2 Novice, 2 Expert Scorers)	Good agreement between all observers was observed (ICC = 0.851) when all three time points were examined. However, interrater reliability differed across timepoints. The best agreement was achieved for orbital tightening, and the poorest agreement for nose and cheek bulge, and this depended on the observers’ experience levels. In general, experienced observers produced scores of higher consistency when compared to inexperienced.
[20]	7	Inter-rater reliability was high as assessed by intra-class correlation coefficient (ICC average = 0.90). When high-definition video cameras were used, over 97% of pain versus no-pain images were categorised correctly.
[59]	6	ICC and Cronbach’s alpha values were low (ICC average < 0.7, α < 0.8). This resulted from large intra-coder variability for three of the coders. Therefore, only the results of the coders with low variability were used in data presentation (updated metrics not reported).
[60]	2	Correlation coefficients ranged between 0.89 and 0.92.
[63]	4	There was high inter-observer consistency, with ICC values ranging from 0.75–0.84.
[64]	6 Novice and 6 Expert Scorers	The α values for experts and novices were high (0.88 to 0.94; 0.78 to 0.87 respectively). Agreement between novices and experts was generally good (ICC ranging from 0.7 to 0.84 across the timepoints).
[65]	2	Moderate to high inter-rater correlation (r = 0.64, *p* < 0.001). Group data from one rater compared to the other were almost identical.
[67]	2	High inter-rater consistency with Cronbach’s alpha of 0.89.
**Inter-Laboratory Variability**		
[46]	3	Median MGS scores were significantly different at a number of timepoints between the 3 laboratories. They were however qualitatively similar i.e., direction of effect.

## Data Availability

The data presented in this study are available on request from the corresponding author.

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
