# Peer review of "Methods Used and Application of the Mouse Grimace Scale in Biomedical Research 10 Years on: A Scoping Review"

_animals, 2021, doi:10.3390/ani11030673_

Round 1

Reviewer 1 Report

This is a really interesting paper and you have done all to provide some really useful and interesting figures in the work - well done! There are only minor formatting errors, but otherwise I commend you on a well written paper that was very nice to read.

Author Response

Thankyou for your review. We hope that formatting errors have been fixed now or will be picked up at the editing stage.

Reviewer 2 Report

This paper reviews reports published in recent 10 years relating to the mouse grimace scale (MGS) after it has been proposed to evaluate degree of the pain that mice suffered in biomedical researches. The contents are valuable for researchers so that it would be eligible to be published. However, re-consider the following minor points.

  1. Ln 109. Spell out "JBI".
  2. Ln 181, Table 1. Please use correct nomenclatures of mouse strains for the following references, although the original referred publications use incorrect them. Refs. 60, 65, and 71.
  3. Ln 235. Delete a space in "black- coloured".

Author Response

Thankyou for your review report and for these suggestions. We have amended as below.

  1. Ln 109. Spell out "JBI"- My coauthor who works for JBI suggests this spelling out would be inappropriate since the name of the Institute has officially been changed to 'JBI' and I guess this forms part of their branding so we woudl prefer to leave as is.  We  do understand the reason for your suggestion though.  
  2. Ln 181, Table 1. Please use correct nomenclatures of mouse strains for the following references, although the original referred publications use incorrect them. Refs. 60, 65, and 71

         Thankyou for alerting us to this error that we didn't pick up. We have now            corrected the nomenclatures in these references. 

  1. Ln 235. Delete a space in "black- coloured"- Thankyou- this has been corrected. 

Reviewer 3 Report

The Review of Whittaker et al. describes methods used and application of the Mouse Grimace Scale during the last 10 years.  I have no comments on this very detailed and carefully crafted review and I would like to congratulate the authors for this excellent work.

I just noticed a little something in the formatting: In the introduction, the reference is given after the dot (for example page 3, line 110) and sometimes before the dot (for example page 3, line 98).

Author Response

Thankyou for alerting us to the formatting errors. The references have all been amended to sit before the punctuation.